# Sex Differences and the Relationship Between Athlete Anthropometrics and Long Jump Performance at National Elite Level

**DOI:** 10.3390/jfmk10010078

**Published:** 2025-02-26

**Authors:** Godwin Chinedu Uzomba, Philip X. Fuchs, Cristina Cortis, Andrea Fusco

**Affiliations:** 1Department of Anatomy, Faculty of Basic Medical Sciences, College of Medical Sciences, Alex Ekwueme Federal University Ndufu-Alike, Abakaliki 84001, Nigeria; uzomba.godwin@funai.edu.ng; 2Department of Physical Education and Sport Sciences, National Taiwan Normal University, Taipei 116, Taiwan; 3Department of Human Sciences, Society and Health, University of Cassino e Lazio Meridionale, 03043 Cassino, Italy; cristina.cortis@unicas.it; 4Department of Medicine and Aging Sciences, University “G. D’Annunzio” of Chieti-Pescara, 66013 Chieti, Italy; andrea.fusco@unich.it

**Keywords:** athletes, gender, anthropometry, performance, regression analysis

## Abstract

Objectives: Anthropometric characteristics influence performance and development in athletic activities such as long jumping. This study aimed to analyze sex differences in anthropometrics among high-level long jumpers and investigate the relationship between anthropometrics and long jump distance. Methods: During the national championships, body height, mass, segment lengths, and circumferences of 39 male and 22 female competitors were obtained via a stadiometer, weight scale, and non-stretchable tape. Officials measured jump distances during the competition. ANOVA, correlation, and stepwise-forward regression analysis were conducted at a significance level of *p* < 0.05. The half-split method was used to cross-validate the final regression model. Results: Height, mass, and more than 50% of the measured segment lengths and circumferences differed between sexes (η^2^ = 0.053–0.422, *p* < 0.05). Jump distance correlated with sex, mass, height, arm span, shank and leg length, and upper arm and chest circumference (*r* = 0.264–0.686, *p* < 0.05). The final regression model identified sex and chest circumference as predictors of jump distance (adjusted *R*^2^ = 0.519, *p* < 0.001). Conclusions: This study enhances the understanding of key anthropometric features influencing long jump performance at an elite level. Recognizing the importance of these characteristics has practical implications for talent identification, athlete assessment, and strength program development.

## 1. Introduction

Jumping is an acyclic movement that requires lower-limb muscular abilities essential for various sports [1,2]. The long jump consists of four distinct phases: run-up, touchdown, flight, and landing [3]. During the run-up, the athlete accelerates until reaching the take-off board [4]. Touchdown occurs when the dominant foot contacts the board, initiating push-off to propel the body forward and upward. After take-off, the athlete enters the flight phase, where the body trajectory is no longer influenced. Landing is prepared through posture adjustments and begins with the first ground contact after take-off.

Running speed during the run-up is a key determinant of long jump performance [5]. Accordingly, elite long jumpers exhibit superior sprinting ability and greater trunk and lower limb musculature (e.g., semitendinosus and rectus femoris) [6,7,8,9,10,11]. Muscular strength and technical coordination are also important during touchdown to optimize propulsion [12]. Lower-limb muscles generate ground reaction forces (GRF), propelling the athlete forward and upward [4]. Upward propulsion, combined with the vertical center of mass (CoM) position at take-off, influences flight duration and, consequently, the horizontal distance traveled [13].

Anthropometric characteristics provide valuable insights into an athlete’s body composition [14,15,16] and its relationship with performance factors. A previous study has observed positive correlations between lower limb length, body height, and vertical jump height [17]. Longer legs have also been linked to greater power generation during push-off in long jumps [18] and enhanced running speed due to increased step length. Additionally, segment circumferences may indicate greater muscle cross-sectional area and strength, which are critical for run-up speed and propulsion [4,6].

Beyond lower-body characteristics, upper-body anthropometrics also influence long jump performance. Longer arms generate greater momentum, aiding in balance and coordination during the run-up and touchdown [19]. Arm swing momentum contributes to landing preparation through powerful shoulder movements [20] and helps achieve an optimal landing posture [21,22]. Collectively, upper and lower body anthropometrics play a key role in jump performance.

While such effects of anthropometrics have been reported in the standing long jump [23], limited research exists for competitive long jumping. Current literature primarily provides theoretical descriptions, mechanical models [24], and experimental studies involving male athletes [25,26]. The mechanical model [24] included CoM height at take-off as the only variable related to anthropometrics (i.e., length of lower limbs and distribution of segment masses). This finding suggested the beneficial effects of longer lower limbs and greater segment masses in the upper body. The experimental studies [25,26] involved only male participants, neglecting well-known sex differences in anthropometrics [27]. For example, males are generally taller than females [27], leading to a higher CoM position. Furthermore, sex differences have been reported in arm usage and strategies for transitioning horizontal to vertical velocity during sport-specific jumps [28], which may influence long jump performance differently.

Understanding sex differences in anthropometrics [27] with sex-specific effects on performance [28] is crucial for optimizing training. If training and monitoring strategies are based on performance factors relevant to only one sex, they may overlook the key aspects of the other. Addressing these gaps contributes to understanding anthropometrics and may enhance talent identification, performance monitoring, and training adjustments.

This study aimed to investigate the relationship between anthropometrics and long jump distance, analyze sex differences in anthropometric characteristics, and develop a model that accounts for sex and predicts long jump distance based on anthropometrics. We hypothesized (1) the presence of significant sex differences in anthropometrics and (2) significant relationships between anthropometric variables and competitive long jump distance among national elite athletes.

## 2. Methods

### 2.1. Participants

The sample included all finalists of the national championships (*n* = 61), comprising 39 men (age: 22.72 ± 4.12 years, body height: 1.77 ± 0.07 m, and mass: 72.96 ± 5.61 kg) and 22 women (age: 21.64 ± 4.09 years, body height: 1.68 ± 0.07 m, and mass: 60.53 ± 9.24 kg). All athletes were classified as national elite by the Athletics Federation of Nigeria (AFN). A-priori power analysis via G*Power 3.1.9.2 indicated that a sample size of 68 would be required to detect moderate effects with the desired statistical power of 1 − β = 0.80 at α = 0.05 for the statistical test with the highest sample size demand (i.e., linear multiple regression with two predictors) [29]. While the study was slightly underpowered for regression analysis, it was adequately powered for all other analyses. The local ethics committee (UPH/CEREMAD/REC/04) approved the study in accordance with the Declaration of Helsinki (2013). Informed consent was obtained from all athletes and their coaches. All participants were healthy and reported no injuries at the time of data collection.

### 2.2. Research Design

This observational study employed a cross-sectional design and collected data in a competitive setting. Two types of data were obtained through different methods: First, individual information and anthropometrics were collected by researchers before the competition; second, long jump distances were measured by officials during the final event of the national championships.

### 2.3. Data Collection

Anthropometric characteristics were obtained via a stadiometer featuring a weight scale (model RGZ-160, Jiangsu Suhong Medical Instruments Co., Ltd., Shanghai, China) and non-stretchable anthropometric tape (see Figure 1). Measurements included body height, mass, arm span, and segment lengths and circumferences (arm, chest, waist, hip, and thigh). From these data, body mass index (BMI) and segmental ratios (one segment divided by another) were calculated. Landmarks were defined at 19 locations on the lower limbs (including the hips), 12 on the upper limbs (including the shoulders), 7 on the trunk, and 1 on the top of the head (see Figure 2 in Wang et al. [30]).

As the present study was conducted during the national championships (see Figure 2), researchers did not intervene in athletes’ warm-up routines. Each athlete followed their individually optimized pre-competition warm-up, which typically included cardiovascular and muscular activation, stretching, and as many familiarization trials as desired. A standard long jump pit with proper markings and a take-off board was used, following official competition norms. In line with competition rules, athletes attempted to achieve the longest possible jump. No additional instructions were given. Each athlete performed one jump per round, with all athletes jumping sequentially. A total of six rounds were completed, yielding six jumps per athlete. Officials measured jump distances via measuring tape in accordance with official regulations. The longest recorded jump for each athlete was analyzed in this study.

### 2.4. Statistical Analysis

Microsoft Office 2007 (Microsoft Corporation, Redmond, WA, USA) and SPSS version 23.0 (SPSS Inc., Chicago, IL, USA) were used to organize and analyze data. Descriptive statistics were presented as mean ± standard deviation (SD) and 95% confidence intervals (CI). The Shapiro-Wilk test assessed normal distribution, guiding the selection of appropriate statistical tests. Spearman correlation analysis examined the relationship between jump distance and anthropometric characteristics. Differences between females and males were tested via multivariate analysis of variance (MANOVA). Prior to regression analysis, collinearity among independent variables was excluded. A multiple stepwise-forward regression analysis was conducted, incorporating sex as a control variable. Effect sizes were reported as correlation coefficient (*r*) for correlation, partial eta squared (η^2^) for group differences, and coefficient of determination (adjusted *R*^2^) for regression analysis. Statistical significance was set at *p* < 0.05.

Post-hoc power analysis indicated an achieved statistical power of 1 − β = 1 for the total sample and 1 − β = 0.999 for half the sample. Given these outcomes, a half-split cross-validation of the regression model was conducted. The sample was randomly divided into two equally sized groups, maintaining the sex distribution of the entire dataset. The regression analysis was first applied to one group, and the resulting regression equation was tested on the second group. A two-way ANOVA with repeated measures examined interactions between groups and differences between measured and estimated jump distances. Additionally, Lin’s concordance correlation coefficient (*CCC*), with 95% confidence intervals, was calculated separately for each group to assess the exact agreement between measured and estimated jump distances [31]. Comparing *CCC* values between groups helped determine whether the regression model maintained its predictive accuracy in different samples.

## 3. Results

Descriptive results are presented in Table 1. Four of the 27 analyzed variables correlated with jump distance (Table 2). No significant correlations were found for segment lengths or ratios. MANOVA confirmed sex differences across all individual characteristics (*F*_27,33_ = 12.316, *p* < 0.001, η^2^ = 0.910, 1 − β = 1). Effects within single factors are detailed in Table 2.

Collinearity analysis revealed significant correlations between sex and mass (*r* = −0.599, *p* < 0.001), body height (*r* = −0.541, *p* < 0.001), arm span (*r* = −0.512, *p* < 0.001), and shank length (*r* = −0.283, *p* = 0.027). Additionally, chest circumference correlated with leg length (*r* = 0.415, *p* = 0.001) and arm circumference (*r* = 0.423, *p* = 0.001). After excluding collinear variables, the final regression model (adjusted *R*^2^ = 0.519, *p* < 0.001) produced the following equation:*Jump distance* (m) = 1.001 *sex* + 0.025 *chest circumference* (cm) + 5.446 (sex: 1 = males, 0 = females).

For the half-split cross-validation, agreement between measured and estimated jump distances was *CCC* = 0.669 (0.441–0.816) and *CCC* = 0.683 (0.469–0.822) for the two groups. A two-way ANOVA with repeated measures found no interaction between the groups and the difference between measured and estimated jump distances (*F*_1,59_ = 0.263, *p* = 0.610, η^2^ = 0.004, 1 − β = 0.080).

## 4. Discussion

This observational performance analysis investigated sex differences in anthropometrics and their relationship with long jump performance in a competitive setting. Hypothesis 1 was confirmed, as sex differences were observed in body height, mass, BMI, most segment lengths, and half of the segment circumferences. Hypothesis 2 was also supported, with significant correlations found between jump distance and sex, body height, mass, three-segment lengths, and two-segment circumferences. The main finding of this study was the differences in multiple anthropometric characteristics and jump distance between females and males, aligning with previous research [27]. The strongest correlation with jump distance was found for sex, followed by body height, body mass, BMI, arm span, shank and total leg length, and upper arm and chest circumference. This study further developed a regression model that identified key individual and anthropometric predictors of jump distance.

The positive correlation between jump distance and body mass was largely attributable to sex differences, as confirmed by collinearity between body mass and sex. While previous studies have suggested that higher body mass negatively influences explosive movements [32], this interpretation is primarily applicable to general populations. In elite athletes with low body fat percentages, increased body mass may instead reflect greater muscle mass, which enhances force production and propulsion.

Body height was another key determinant of long jump distance, also correlating with sex. A previous study has linked greater height and leg length to an increased range of motion and potential energy in vertical jumping [19]. In addition, a higher CoM position at take-off, associated with taller athletes, is advantageous in long jump performance [27]. Longer legs further contribute to a higher CoM position at take-off [33,34], supporting the observed positive correlation between jump distance, shank length, and total leg length. In vertical jumping, longer legs provide mechanical advantages, potentially contributing to greater force production [35]. These findings reinforce the biomechanical benefits of body height and lower limb length in long jump performance.

Upper limb characteristics also play an important role in long jumping [3,22]. In the present study, arm span and upper arm circumference correlated with jump distance, likely due to their impact on rotational momentum during arm swing. A greater arm span allows for increased angular momentum, which influences lower body motion [3,22] and optimizes landing posture [20].

An unexpected but noteworthy finding was that chest circumference emerged as the most valuable anthropometric predictor of jump performance alongside sex. The final regression model, incorporating only sex and chest circumference, explained 52% of the variance in jump distance. While previous studies have highlighted body height, BMI, and body segment ratios as important predictors [28,36,37], the current results suggested that these variables provide redundant information already reflected by sex. However, their exclusion from the regression model does not diminish their practical relevance, as they remain systematically linked to sex-related performance differences.

The strong association between chest circumference and jump distance may reflect the role of upper-body muscle mass. Chest circumference is influenced by muscles such as the pectoralis major and latissimus dorsi, which contribute to shoulder movement [22]. Dynamic arm movement is important for sprint running at maximum speed [38]; since take-off velocity is a key determinant of long jump distance [5], greater upper-body muscle mass may enhance sprinting and long jump performance.

Collinearity was tested and excluded before conducting stepwise-forward regression to avoid redundancy among predictors. This process removed two lower limb variables (leg and shank length) that correlated with jump distance but did not significantly improve the model beyond stronger predictors. While no lower limb variables were included in the final regression model, this does not imply their irrelevance for long jump performance. The lower limbs remain fundamental for generating propulsion and achieving optimal take-off mechanics [3,7,10,13].

The half-split cross-validation yielded 66.9–68.3% agreement between the measured and estimated jump distance. While not sufficient for highly precise individual predictions, it is noteworthy that a model with only two predictors—one being a categorical variable (sex)—accounted for the majority of performance variability. The consistency between the two subsamples and the absence of interaction effects further supported the model’s reliability, indicating the transferability of the findings to similar populations.

## 5. Practical Implications

The current findings contributed to understanding the key anthropometric factors in elite national-level long jumping. Sex, body height, body mass, arm span, shank and leg length, and upper arm and chest circumferences were identified as influential for jump distance. The significance of chest circumference, irrespective of sex, suggested that a stronger upper body may enhance jump performance by arm movement during the run-up, maintaining body alignment, facilitating effective force transfer from the lower to the upper body, and optimizing take-off speed. Therefore, coaches and athletes may consider emphasizing upper body training in strength programs to enhance upper body strength, which is associated with chest circumference. Additionally, coaches may tailor strength programs based on an athlete’s anthropometric profile, addressing individual weaknesses. Beyond training implications, understanding sex differences and anthropometric performance determinants can support athlete monitoring and talent scouting.

### Limitations

The study focused on the anthropometric factors and did not examine other potential influences on long jump performance, such as technical jumping skills, physiological attributes, or mental state. While anthropometrics play a role, other factors may also be important. The findings were derived from a sufficiently large sample of high-level long jumpers and may not be generalizable to other sports and skill levels. Although the current results identify key determinants in national elite athletes, these factors may not be the most decisive at different skill levels or distinguish between them. Future research should explore the effects of technique, physiology, and mental abilities across broader populations and sports to improve the generalizability of the findings.

## 6. Conclusions

This study provided rare insights into the anthropometric profiles of all competitors at the national championships, identifying sex differences in over half of the investigated variables, many of which correlated with long jump distance. The findings highlighted the role of anthropometric characteristics in long jumping, particularly emphasizing the relevance of chest circumference, which may be undervalued in current long jump practices. This study offered foundational knowledge for strength and conditioning coaches and sports scientists, with applications in talent scouting, performance monitoring, and individualized training program design. The results supported the importance of a well-developed physique, especially chest circumference, in achieving great long jump distances for both female and male elite athletes.

## Figures and Tables

**Figure 1 jfmk-10-00078-f001:**
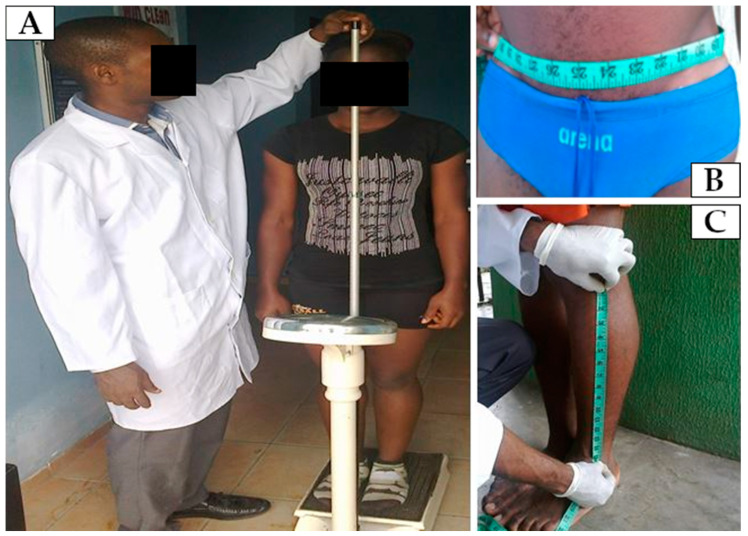
Photos of the procedure and instruments used for anthropometric data collection, including a stadiometer with a weight scale for measuring body height and weight (**A**) and non-stretchable anthropometric tapes for measuring circumferences (**B**) and segment lengths (**C**).

**Figure 2 jfmk-10-00078-f002:**
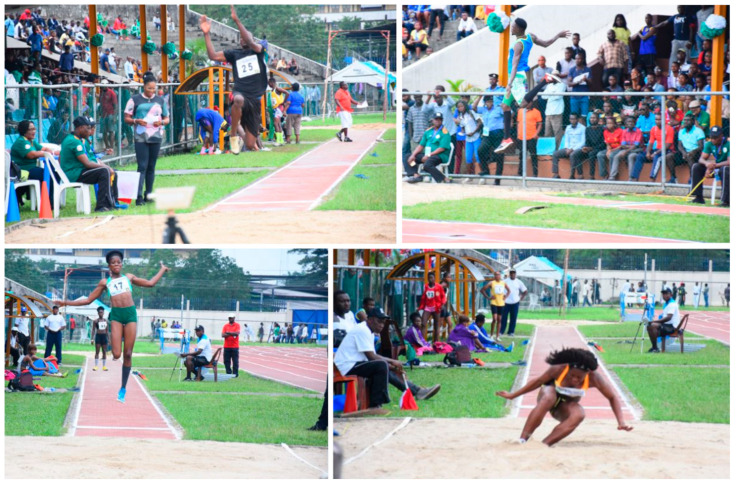
Photos of the long jump competition where officials measured jump distance.

**Table 1 jfmk-10-00078-t001:** Anthropometrics and descriptive results as mean ± standard deviation (SD) and confidence intervals (95% CI) for males and females.

		Males		Females	
	Variable	Mean ± SD	95% CI	Mean ± SD	95% CI
General	Age (years)	22.7 ± 4.1	21.0–24.4	21.6 ± 4.1	19.9–23.3
	Mass (kg)	73.0 ± 5.6	71.7–74.3	60.5 ± 9.2	58.2–62.8
	Height (m)	1.77 ± 0.07	1.67–1.87	1.68 ± 0.07	1.58–1.78
	Body-mass index (kg/m^2^)	23.1 ± 2.1	22.2–24.0	21.3 ± 2.5	20.3–22.3
Segment lengths	Arm span (cm)	188.6 ± 11.9	186.9–190.3	176.8 ± 7.0	175.7–177.8
	Upper arm (cm)	36.0 ± 4.1	34.8–37.3	34.7 ± 2.5	33.8–35.5
	Forearm (cm)	31.5 ± 3.0	30.4–32.6	29.8 ± 2.5	29.9–30.8
	Thigh (cm)	60.7 ± 5.9	59.2–62.2	60.0 ± 6.0	58.5–61.5
	Shank (cm)	45.2 ± 4.4	43.9–46.5	42.8 ± 3.1	41.9–43.7
	Leg (cm)	97.3 ± 6.4	96.0–98.6	95.1 ± 7.0	93.7–95.1
	Torso (cm)	54.7 ± 3.7	53.7–55.7	51.7 ± 5.3	50.3–53.1
Circumferences	Arm (cm)	29.0 ± 3.7	27.7–30.3	27.1 ± 3.1	25.9–28.3
	Chest (cm)	83.4 ± 11.9	80.8–86.0	83.3 ± 5.4	82.1–84.5
	Waist (cm)	78.5 ± 6.3	77.1–79.9	75.3 ± 7.2	73.7–76.9
	Hip (cm)	81.9 ± 7.2	80.3–83.5	77.2 ± 7.7	75.5–78.9
	Thigh (cm)	54.4 ± 5.0	53.1–55.7	54.4 ± 4.6	53.0–55.8
	Shank (cm)	36.6 ± 3.0	35.6–37.6	35.2 ± 3.2	34.1–36.3
Ratios	Waist/hip (circumferences)	0.96 ± 0.03	0.95–0.97	0.98 ± 0.01	0.98–0.98
	Shank/thigh (lengths)	0.75 ± 0.08	0.72–0.78	0.72 ± 0.89	0.35–1.09
	Shank/arm (lengths)	1.27 ± 0.18	1.21–1.33	1.24 ± 0.12	1.19–1.29
	Shank/forearm (lengths)	1.44 ± 0.14	1.40–1.48	1.45 ± 0.18	1.37–1.53
	Shank/torso (lengths)	0.83 ± 0.09	0.80–0.86	0.84 ± 0.13	0.79–0.89
	Shank/(arm + forearm) (lengths)	0.67 ± 0.07	0.65–0.69	0.67 ± 0.07	0.64–0.70
	Shank/(arm + torso) (lengths)	0.51 ± 0.06	0.50–0.53	0.51 ± 0.06	0.48–0.54
	Thigh/(arm + forearm) (lengths)	0.90 ± 0.09	0.87–0.93	0.93 ± 0.10	0.89–0.89
	Thigh/(arm + torso) (lengths)	0.67 ± 0.07	0.65–0.69	0.70 ± 0.07	0.67–0.73
	Jump distance (m)	6.55 ± 0.62	6.36–6.74	5.55 ± 0.49	5.36–5.74

**Table 2 jfmk-10-00078-t002:** Correlation among jump distance, anthropometrics, and sex differences in anthropometrics. Significant effects were marked in bold.

	Variables	Correlation (Jump Distance)		Difference (Sex)		
		*r*	*p*	η^2^	*p*	1 − β
General	Sex	**0.686**	<0.001	-	-	
	Age	0.097	0.459	0.016	0.327	0.163
	Mass	**0.338**	0.008	**0.421**	<0.001	1
	Height	**0.375**	0.003	**0.297**	<0.001	0.998
	Body-mass index	0.182	0.160	**0.148**	0.002	0.883
Segment lengths	Arm span	**0.318**	0.012	**0.233**	<0.001	0.986
	Upper arm	0.200	0.123	0.031	0.172	0.274
	Forearm	0.214	0.098	**0.083**	0.024	0.624
	Thigh	0.183	0.157	0.004	0.619	0.078
	Shank	**0.300**	0.019	**0.078**	0.029	0.595
	Leg	**0.264**	0.040	0.034	0.157	0.292
	Torso	0.137	0.291	**0.105**	0.011	0.734
Circumferences	Upper arm	**0.301**	0.018	**0.069**	0.041	0.537
	Chest	**0.353**	0.005	<0.001	0.958	0.050
	Waist	−0.167	0.197	**0.053**	0.075	0.431
	Hip	−0.123	0.345	**0.087**	0.021	0.645
	Thigh	0.130	0.318	<0.001	0.982	0.050
	Shank	−0.020	0.878	0.050	0.084	0.410
Ratios	Waist/hip (circumferences)	−0.228	0.078	**0.067**	0.044	0.526
	Shank/thigh (lengths)	0.055	0.676	0.027	0.208	0.240
	Shank/arm (lengths)	−0.010	0.938	0.007	0.532	0.095
	Shank/forearm (lengths)	0.009	0.948	<0.001	0.912	0.051
	Shank/torso (lengths)	0.079	0.547	0.005	0.595	0.082
	Shank/(arm + forearm) (lengths)	−0.017	0.897	<0.001	0.891	0.052
	Shank/(arm + torso) (lengths)	0.033	0.799	0.003	0.653	0.073
	Thigh/(arm + forearm) (lengths)	−0.036	0.785	0.030	0.184	0.262
	Thigh/(arm + torso) (lengths)	0.014	0.914	0.039	0.127	0.331
	Jump distance	-	-	**0.422**	<0.001	1

## Data Availability

Data are available on request.

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
