# Peer review of "Sex Differences and the Relationship Between Athlete Anthropometrics and Long Jump Performance at National Elite Level"

_jfmk, 2025, doi:10.3390/jfmk10010078_

Round 1

Reviewer 1 Report

Comments and Suggestions for Authors

Review - Sex  Differences and the Role of Anthropometrics in Achieving  Great Long Jump Performance at National Elite Level

General Comments

Since the authors have selected an interesting research topic, their work should be acknowledged. The present study contributes to the understanding of relevant anthropometrics in competitive long jumping at an elite national level, for both male and female athletes. However, the manuscript incurs in some concepts and methodological  inaccuracies that should be revised along the manuscript. The authors were encouraged to include images from the data collection and treatment. So, with these appointments the merit  of the present study  is compromised. I suggest the authors include all of my comments on the manuscript. After, I would like to review it again.

 I hope that the minor comments can help the authors.  

Specific Comments

Title

Suggestion: The relationship between athlete characteristics and anthropometrical parameters to  long jump performance.

Abstract

L18 to L19 – The background should be modified for anthropometrics and long jump performance.

L19 – the present study aims to analyze.    

L22 to 23 – A brief explanation about the experimental protocol should be included.

L29 to L30 – Insert the effect size and the p value.

Introduction

The research question of the present study is not clear. Justify, based on specialized literature, why it is relavant to assess the differences on long jump performance between males and females? The authors, in none moment in this section described about  the deterministic models, for example, The Jim Hay method applied to high performance in  long jump movement. In those models, the author describes the some anthropometrical variables determinant in long jump performance. It will be interesting the authors include some information about this assumption for the reader understand the research problem of this paper.  What are the main expected differences between both sexes?  Explaning this for the reader, it is also possible to raise a hypothesis in this study.  

Methods

L90 to L95 – information about the main training characteristics should be included 

L117 – See my comments in L90.

L116 to L123 – Which part of the event the authors collected the data? Semi – finals, Finals...It is not clear here.

In this section figures, images are strongly encouraged. Figures about the body marks on the athlete´s body. Figures about how the anthropometric data was assessed... Figure about the instruments used...

L146 to 148 – Insert the intervals between the s Lin’s concordance correlation coefficient (CCC)

Results

L152 to L166 – Before to run the regression analysis, did authors checked the autocorrelated variables? It is very strange do not entry lower limbs variables in the regression model.

Discussion

L181 – A Brief discussion about the main results should be included.

L205 to L 210 – Insert a pair of references here.

L217 to L220 – I´m not convinced of this argument. See my comment in L152 to 156.

L224 to L227 – Quote a pair of reference here.

A para about the methodology applied in the present study should be included.   

L257 to L264 – For each limitation the authors should present a solution.

Author Response

Reviewer 1

General Comments

Since the authors have selected an interesting research topic, their work should be acknowledged. The present study contributes to the understanding of relevant anthropometrics in competitive long jumping at an elite national level, for both male and female athletes. However, the manuscript incurs in some concepts and methodological inaccuracies that should be revised along the manuscript. The authors were encouraged to include images from the data collection and treatment. So, with these appointments the merit of the present study is compromised. I suggest the authors include all of my comments on the manuscript. After, I would like to review it again.

 I hope that the minor comments can help the authors.  

Response: The authors appreciate the reviewer’s recommendations that helped further strengthening our manuscript.

Specific Comments

Title

Suggestion: The relationship between athlete characteristics and anthropometrical parameters to long jump performance.

Response: Thank you for the suggestion. We modified the title, integrating the suggestion in a condensed form into the original version. The original aspect of specifically pointing out the sex differences was considered valuable because it is an essential part of the work with a distinct analysis that would not be reflected in “relationship” alone. The modified title reflects both aspects (original and reviewer suggestion).

Abstract

L18 to L19 – The background should be modified for anthropometrics and long jump performance.

Response: As per reviewer’s comment, we modified the first sentence (L19), now referring to anthropometrics and specifically long jump performance.

L19 – the present study aims to analyze.    

Response: As per reviewer’s suggestion, we incorporated “present” and adjusted “to analyze”. We maintained the past tense of “to aim” (aimed) as the event of defining the aim of the study happened in the past. If the present tense is desired as a standard in JFKM, we will gladly align with the editor’s requests.

L22 to 23 – A brief explanation about the experimental protocol should be included.

Response: We rearranged (L22) and added the specific tools used to obtain anthropometrics (L23-24). The protocol consisting of measurement of anthropometric data and jump length is fully described, now with additional detail about measurement tools.

L29 to L30 – Insert the effect size and the p value.

Response: The effect size has been presented as η² (ANOVA), r (correlation), and adjusted R² (regression). The p values have been indicated as *. For clarification, we changed * to the specific p values with the concrete numbers for all results.

Introduction

The research question of the present study is not clear. Justify, based on specialized literature, why it is relavant to assess the differences on long jump performance between males and females? The authors, in none moment in this section described about the deterministic models, for example, The Jim Hay method applied to high performance in long jump movement. In those models, the author describes the some anthropometrical variables determinant in long jump performance. It will be interesting the authors include some information about this assumption for the reader understand the research problem of this paper.  What are the main expected differences between both sexes?  Explaning this for the reader, it is also possible to raise a hypothesis in this study.  

Response:

  1. Reviewer: Why are sex differences in performance relevant?

Response: The relevance of investigating sex differences has been addressed in the fifth paragraph (L73-97). We added a new description (L86-87 and L88-93) to clarify and strengthen its relevance.

  1. Reviewer: Describe anthropometrical variables in deterministic models.

Response: The deterministic model we found and referred to was Zhiguo Pan’s model (doi: 10.19026/rjaset.5.4909). The only variable related to anthropometrics pointed out by Pan was the height of the center of gravity. We added a description to L77-81.

  1. Reviewer: What are expected main effects between sexes?

Response: One expected effect resulting from the abovementioned model’s findings was integrated in the new addition at L79-81. Furthermore, we added concrete information about sex differences in anthropometrics (L83-84) and performance determinants (L85-86) supported by specific literature. Based on this information, we added another statement (L86-87) on expected findings for the present study based on previous findings.

  1. Reviewer: Add a hypothesis.

Response: A hypothesis has been added to the end of the introduction section (L101-103).

Methods

L90 to L95 – information about the main training characteristics should be included 

Response: We understand the reviewer’s interest in the athletes’ training characteristics. However, we do not have this information, and we could not have access to this information even if we intended to collect it at the time of data collection because the study participants were the competitors at the national championships. They were not “our” athletes. So, we do not know how they train, and – certainly – their coaches would not have agreed to share their personal training practices with us for publication. All available information (age, anthropometrics, and competition performance) was provided in the manuscript.

L117 – See my comments in L90.

Response: Thank you. We added more detailed information about the procedure (L135-139).

L116 to L123 – Which part of the event the authors collected the data? Semi – finals, Finals...It is not clear here.

Response: Thank you for the question. We added a clarification at the suggested location (L144-146) and also in the participants section (L106), stating that data were collected during the entire competition, which was the final event of the national championships.

In this section figures, images are strongly encouraged. Figures about the body marks on the athlete´s body. Figures about how the anthropometric data was assessed... Figure about the instruments used...

Response: First, we added information about the landmarks and referred to the specific figure in the original publication where the detailed definitions can be retrieved. This modification was added at the location where we clarified the marker specifications (L131-134). Second, as per reviewer’s suggestion, we designed a figure (Figure 1, L147) that combines several pictures displaying the instruments and procedures during data collection.

L146 to 148 – Insert the intervals between the s Lin’s concordance correlation coefficient (CCC)

Response: We added 95% confidence intervals at this location in the methods section (L175) and also to the results section accordingly (L192).

Results

L152 to L166 – Before to run the regression analysis, did authors checked the autocorrelated variables? It is very strange do not entry lower limbs variables in the regression model.

Response: We assume the reviewer means co-linearity (/co-correlation). Yes, as it is common practice, we tested co-linearity (/co-correlation) and removed it from further analysis (described in the statistics section, L160). Detected co-linearity was reported in the results section (L185-188). We share the reviewer’s surprise that no lower limb variables contributed to the final regression model. But in fact, leg and shank length did correlate with performance but also co-correlated with stronger predictors (see Table 2 and L185-188). So, yes, co-linearity was the reason why no lower limb variable entered the final model.

Discussion

L181 – A Brief discussion about the main results should be included.

Response: Main results have been discussed in the first paragraph of the discussion section (L203-211). We modified this part by rearranging and rephrasing in order to clarify that these statements address the main findings.

L205 to L 210 – Insert a pair of references here.

Response: Thank you for the suggestion. We added three references supporting the statements in this paragraph (L234, 239, and 240).

L217 to L220 – I´m not convinced of this argument. See my comment in L152 to 156.

Response: We hope that our response to the previous comment solves the confusion. The statement mentioned here (new lines: L247-250; “However, other […] with jump distance”) purely reflects the current results. We modified the phrase to clarify that this statement is not an argument but an observation based on the current study’s data.

L224 to L227 – Quote a pair of reference here.

Response: As per reviewer’s suggestion, we added a reference at L257.

A para about the methodology applied in the present study should be included.   

Response: We added a paragraph (L263-271) addressing the impact of the applied methods. In particular, the new paragraph focuses on what we considered a critical part of the study (regression analysis and the fact that no lower limb variable was included in the final model) that could easily cause misunderstanding. Therefore, we hope that this paragraph not only meets the reviewer’s original expectation but also strengthens the interpretation of the current results.  

L257 to L264 – For each limitation the authors should present a solution.

Response: Thank you. We added a statement at the end of the section (L306-307) with a suggestion for future research addressing each limitation.

Reviewer 2 Report

Comments and Suggestions for Authors

It is an interesting study on performance variables in the long jump

Introduction

This could also indicate that, depending on anthropometric characteristics, there is a better technique in the flight phase (extension, 2-3.5, etc).

These differences, given the title, could be differentiated by sex to identify whether all the variables coincide or whether the male and female categories are different; otherwise, as indicated in one of the variables, it should be said that it is only male (line 76).

Methods

Given what the authors indicate about the sample, it is suggested to add to the title of the study that it is a pilot study, as they themselves recognize that a larger number would be desirable.

I would like to know the reason for not measuring the wingspan, as the importance of the upper limbs has been argued.

In this section, it is worth adding a picture to provide an overview of the measurements and protocol.

Discussion

The results were clearly different. However, there is a lack of practical proposals beyond those that have appeared. For example, the possibilities of advising training and guidelines to be taken into account by coaches, depending on the profile of their athletes, I think that this would enrich the work.

Author Response

Reviewer 2

Comments and Suggestions for Authors

It is an interesting study on performance variables in the long jump

Introduction

This could also indicate that, depending on anthropometric characteristics, there is a better technique in the flight phase (extension, 2-3.5, etc).

These differences, given the title, could be differentiated by sex to identify whether all the variables coincide or whether the male and female categories are different; otherwise, as indicated in one of the variables, it should be said that it is only male (line 76).

Response: Differences in technique during the flight phase are possible but have not been the topic of the present study. The study focused on differences in anthropometrics and their relationships with performance, which was analyzed accordingly. To overcome the mentioned limitation of previous studies investigating only males, the present data included both males and females, and the current analysis controlled for sex. We hope that our response addresses the reviewer’s thoughts. If the reviewer suggests any specific modifications in the manuscript, please specify.

Methods

Given what the authors indicate about the sample, it is suggested to add to the title of the study that it is a pilot study, as they themselves recognize that a larger number would be desirable.

Response: We addressed the sample’s limitation to a specific skill level, not to its size. To clarify this, we modified the wording in the limitation section (L301-302) and added a phrase that addresses applicability due to sample characteristics (L306-307). In fact, the current sample size of n=61 is large compared to most studies in the field. We reported the achieved statistical power (1-β=1) in the manuscript, which strongly corroborates the suitability of the sample size and does not suggest this study as a pilot.

I would like to know the reason for not measuring the wingspan, as the importance of the upper limbs has been argued.

Response: Wingspan was measured and named “Arm span” (see both tables). We agree with the reviewer that arm span may be important, which was confirmed by the currently detected correlation between arm span and jump distance (reported in Table 2).

In this section, it is worth adding a picture to provide an overview of the measurements and protocol.

Response: Thank you for the suggestion. We designed a new figure (Figure 1, L147) displaying pictures of the instruments and various procedures during data collection.

Discussion

The results were clearly different. However, there is a lack of practical proposals beyond those that have appeared. For example, the possibilities of advising training and guidelines to be taken into account by coaches, depending on the profile of their athletes, I think that this would enrich the work.

Response: We modified the Practical Implications section by integrating the reviewer’s suggestion to address the aspect of profile-specific training recommendations for coaches (L292-294) in addition to the previously existing statements at L291-292 and L294-296, which addressed practical implications for coaches.

Round 2

Reviewer 1 Report

Comments and Suggestions for Authors

General Comments – First, I would like to acknowledge the effort made by the authors on the corrections of the present manuscript. The introduction section improved, but there is a huge disequilibrium among the paragraphs. Furthermore now, the section has six paragraphs. In the method section, mainly in about the data collection and treatment. Once again, the figures and images are encouraged to explain the data collection and treatment, mainly in the competitive scenario. The authors included only an image showing the anthropometrical data collection. Include a brief discussion about the hypothesis formulated.  Also, in the discussion a brief discussion about the main results using the specialized literature should be included.  The practical applications are not clear. What the authors mean improve training programs? What kind of training? Specify...

Comments on the Quality of English Language

I strongly encourage the authors to review this manuscript by a native English speaker. 

Author Response

Note from the authors: Thank you for your continued efforts to help us improve the manuscript. As recommended, we carefully reviewed the language and writing throughout the entire manuscript and made necessary revisions.

To clearly differentiate between language edits and changes made in response to other reviewer comments, we used "Track Changes" (all changes) and "Text Highlight Color" (content-related changes in yellow). All line numbers refer to the current version with “Track Changes” set to “All Markup”.

General Comments

First, I would like to acknowledge the effort made by the authors on the corrections of the present manuscript. The introduction section improved, but there is a huge disequilibrium among the paragraphs. Furthermore now, the section has six paragraphs.

Response: Thank you for your positive feedback and for recognizing our efforts to improve the manuscript. We agree with the reviewer’s comment regarding the imbalance in paragraph lengths, particularly the fifth paragraph, which was significantly longer than the others. To address this, we have divided it into two separate paragraphs at L102. As a result, the introduction section now consists of seven paragraphs of more balanced lengths. For comparison, the discussion section contains eight paragraphs of similar length.

In the method section, mainly in about the data collection and treatment. Once again, the figures and images are encouraged to explain the data collection and treatment, mainly in the competitive scenario. The authors included only an image showing the anthropometrical data collection.

Response: Thank you for your valuable feedback. The first paragraph of the data collection section describes the procedure for collecting anthropometric data, while the second paragraph details the procedure during the competition. As per reviewer’s request for images from the competition, we have added a figure (Figure 2) containing multiple pictures of jumps.

Please note that the jumps took place during an official competition, and jump distance was measured by officials following official rules, as described in the second paragraph. For this reason, we did not take pictures of specific moments, such as an official holding the measuring tape.

Additionally, in response to the reviewer’s suggestion, we have provided further details on the competition procedure in the second paragraph (L170 ff.). We hope that the new figure and the additional description address the reviewer’s request.

Include a brief discussion about the hypothesis formulated.

Response: As suggested by the reviewer, we have added statements addressing the hypotheses (L243 ff.).

Also, in the discussion a brief discussion about the main results using the specialized literature should be included.

Response: The main results are discussed in the first paragraph of the discussion (L247-257). To strengthen this section, we have added a reference (L249) that reports similar effects in the most comparable context we could find.

Regarding other parts, we are not aware of additional relevant references that would provide further context or add value to the discussion. If the reviewer has specific suggestions for relevant “specialized literature,” we would appreciate further clarification.

The practical applications are not clear. What the authors mean improve training programs? What kind of training? Specify...

Response: We have made several changes throughout the chapter to improve clarity and have specified the type of training at L360.

I strongly encourage the authors to review this manuscript by a native English speaker.

Response: We had the entire manuscript reviewed and made the necessary revisions accordingly.